# Clinical Features and Outcomes of Japanese Patients with Giant Cell Arteritis: A Comparison with Takayasu Arteritis

**DOI:** 10.3390/jpm13030387

**Published:** 2023-02-22

**Authors:** Shuhei Yoshida, Eiji Suzuki, Yuya Sumichika, Kenji Saito, Haruki Matsumoto, Jumpei Temmoku, Yuya Fujita, Naoki Matsuoka, Tomoyuki Asano, Shuzo Sato, Hiroshi Watanabe, Kiyoshi Migita

**Affiliations:** 1Department of Rheumatology, Fukushima Medical University School of Medicine, 1 Hikarigaoka, Fukushima 960-1295, Fukushima, Japan; 2Department of Rheumatology, Ohta-Nishinouchi Hospital, 2-5-20 Nishinouchi, Koriyama 963-8558, Fukushima, Japan

**Keywords:** giant cell arteritis, Takayasu arteritis, aortitis, LV-GCA, large vessel vasculitis

## Abstract

Background: Giant cell arteritis (GCA) and Takayasu arteritis (TA) are distinct types of large-vessel vasculitis; however, the clinical features of the diseases have some similarities. Limited data are available regarding Japanese patients with GCA and TA. The present study aimed to compare the clinical features and outcomes of Japanese patients with GCA and TA and the effects of large vessel involvement (LVI). Methods: We performed a retrospective cohort study of the patients with GCA (*n* = 15) and TA (*n* = 30) who visited our department from April 2012 to June 2022. Signs and symptoms attributed to the disease, treatment, clinical outcomes, and mortality were recorded using a standardized database. Results: The median age of onset was significantly higher in the GCA group at 24 years (range, 16–72 years) in the TA group and 77 years (range, 57–89 years) in the GCA group (*p* < 0.001). There were no significant differences in survival rates or the cumulative rates of cardiovascular events between the GCA and TA groups. However, relapse-free survival rates were significantly higher in patients with GCA than in patients with TA. Seven of the 15 patients with GCA had large vessel involvement, which did not affect the survival rates. Prednisolone (PSL) doses were significantly decreased after induction therapy in both groups, and the rates of achieving steroid tapering (PSL < 5.0 mg/day) were significantly higher in patients with GCA compared with those in patients with TA. Conclusions: Our study demonstrated no significant difference in the survival rates of Japanese patients with GCA and TA. The relapse-free survival rates were significantly higher in the GCA group than in the TA group. LVI may not be associated with disease relapse or survival rate in Japanese patients with GCA.

## 1. Introduction

Giant cell arteritis (GCA) is a large-vessel vasculitis that predominantly involves the extracranial branches of the carotid arteries in elderly patients. GCA can be accompanied by various complications, including visual disturbances [1]. Additionally, aortic involvement occurs in 40–80% of patients with GCA [2]. It has been reported that GCA is associated with an increased risk of cardiovascular events [3]. Elderly onset GCA and large vessel involvement (LVI) are related to cardiovascular risk factors [4,5]. Glucocorticoids (GCs) are the mainstay treatment for GCA [6,7]. However, high-dose GCs usage and long-term treatment courses are associated with an increased risk of cardiovascular disease [8]. The efficacy of anti-IL-6 receptor tocilizumab in GCA to reduce the accumulated dose of glucocorticoids and the risk of relapse has been demonstrated both in clinical trials and in several studies based on daily clinical practice [9,10,11]. Although GCA and Takayasu arteritis (TA) share some common clinical features, differences in disease manifestations and clinical courses amongst the two respective diseases require more investigation. GCA and TA can be complicated by significant morbidity and mortality due to cardiovascular complications, including aneurysm formation, acute aortic dissection, and stenosis or obstruction [12]. Mortality rates of patients with GCA are comparable to those of the general population; however, an increased risk of potentially life-threatening ischemic or cardiovascular events has been suggested during the therapeutic disease course [13,14,15,16]. In addition, disease flares are frequently observed during tapering GCs dosage or withdrawal [17]. Therefore, an optimal management strategy for GCA has not yet been established.

Since GCA is a rare vasculitis in Asian countries [18], information regarding the differences in clinical outcome between GCA and TA is scarce in real-world. A 2017 Japanese epidemiological study estimated the number of clinically diagnosed TA and GCA patients to be 5320 (95% confidence interval [CI], 4810–5820) and 3200 (95% CI, 2830–3570), respectively [19]. This is an extremely small number in relation to the Japanese population. Because of this, we established a retrospective study of GCA and TA patients seen at our center over a 10-year period.

## 2. Materials and Methods

### 2.1. Patients

We conducted a retrospective observational study at Fukushima Medical University Hospital and Ohta Nishinouchi Hospital. This observational study included 15 consecutive patients with GCA and 30 with TA. All patients were treated at the Department of Rheumatology, Fukushima Medical University Hospital, and Ohta Nishinouchi Hospital from April 2012 to June 2022. This study aimed to examine the clinical characteristics of patients with active GCA and TA. The diagnosis of GCA was based on the American College of Rheumatology (ACR) classification criteria because the Japanese Circulation Society (JCS) has not proposed new diagnostic criteria for GCA [20]. The presence of LVI in GCA was defined as inflammatory findings, or wall thickening of the LV was defined as the aorta and its first branches on imaging studies, including enhanced computed tomography (CT), magnetic resonance angiography (MRA), ultrasonography, or 18F-fluorodeoxyglucose positron emission tomography (PET)/CT. The diagnostic criteria for TA were based on the definition in the “Japanese Guidelines for the Management of Vasculitis Syndrome 2008” [21]. And the follow-up period was defined as the time from diagnosis to death or the latest hospital visit. Patient follow-up will be conducted until June 2022. This study was conducted under the principles of the Declaration of Helsinki. Ethical approval for this study was provided by the Fukushima Medical University Ethics Committee (No. 2020-110).

### 2.2. Clinical Evaluations

Patient medical records were retrospectively reviewed, and clinical data related to the age of onset, sex, symptoms, and physical findings at diagnosis, complications, medications, and surgical interventions were collected. Routine laboratory markers of disease activity, including erythrocyte sedimentation rate (ESR) and C-reactive protein (CRP) level, were collected. Clinical improvement was assessed by evaluating the signs and symptoms of disease activity and the ability to taper prednisone (PSL). Remission was defined as a complete absence of clinical symptoms or signs of disease activity referring to the European Alliance of Associations for Rheumatology (EULAR) recommendation for LVV, updated in 2018 [22]. GCA relapse was based on the consensus definition established in international multicenter clinical trials [23,24,25]. TA relapse was defined as the presence of signs of relapse, as judged by the investigator for at least two of the following four categories according to Kerr’s criteria: systematic features, elevated ESR, features of vascular ischemia or inflammation, and typical angiographic features [26]. Cardiovascular events were defined as LV complications, including chest and back pain associated with arteritis, arterial stenosis, aneurysm, arterial dissection/rupture, and aortic valve stenosis and regurgitation.

### 2.3. Statistical Analysis

Data are presented as medians and ranges for continuous variables and frequencies and percentages for qualitative variables. Fisher’s exact test was used for comparing qualitative variables, and the Mann–Whitney’s U test was used for comparing continuous variables, as appropriate. Time to relapse, death, and cardiovascular events for patients with GCA and TA were estimated using Kaplan–Meier analysis, and log-rank tests were used to compare survival rates between patient groups. All data entry and statistical analyses were performed using SPSS Statistics version 22.0 (IBM, Armonk, NY, USA), except for the relapse rate ratio. The analysis of the relapse rate ratio and its 95% confidence interval (CI) are conducted using R version 4.2.2 (R Core Team (2022). R: A language and environment for statistical computing. R Foundation for Statistical Computing, Vienna, Austria, http://www.R-project.org/ (accessed on 14 February 2023)). In all the analyses, two-sided *p* < 0.05 was considered statistically significant.

## 3. Results

### 3.1. Baseline Characteristics of Patients with GCA and TA

A total of 45 patients were enrolled, including 15 with GCA and 30 with TA; the baseline patient characteristics of GCA and TA and their comparisons are shown in Table 1. Baseline patient characteristics of GCA and TA for which comparisons are not appropriate are shown separately in Table 2 and Table 3, respectively. The median ages at onset were 24 and 77 years for TA and GCA, respectively, and was significantly higher in the GCA group (*p* < 0.001). Similarly, the prevalence of hypertension was significantly higher in patients with GCA than in those with TA (*p* < 0.001). There were no statistically significant differences between the two groups in inflammatory markers and the rate of concomitant use of immunosuppressants. The relapse rate during the entire treatment period and follow-up period was higher in patients with TA (Table 3). The relapse rate ratio for TA group to GCA group was 6.78, indicating that TA group is more likely to relapse than GCA group (95% CI 1.39–162.93, *p* = 0.013). Figure 1 shows the distribution of the affected arteries in patients with GCA and TA. Aortic lesions were found primarily in the carotid artery, descending aorta, and abdominal aorta in patients with GCA. However, the carotid, subclavian, and ascending aorta lesion rates were significantly higher in patients with TA than patients with GCA (Figure 1).

### 3.2. Survival Rates for Patients with GCA and TA

There were no statistically significant differences in the survival rates of the patients with GCA and TA between the two groups (Figure 2). However, comparing the survival rates may not be appropriate because the mean ages of the two groups were significantly different. The causes of death were sudden death of unknown cause (1 GCA), cerebral hemorrhage (1 GCA), and rupture of the ascending aorta (1 TA).

### 3.3. Relapse-Free Survival Rates for Patients with GCA and TA

All patients in the GCA and TA groups achieved remission after initial remission induction therapy with or without immunosuppressive agents. During the entire treatment period, relapse occurred in 1 patient with GCA and in 16 patients with TA. Relapses were accurately assessed according to the consensus definition of GCA relapse [23,24,25] and Kerr’s criteria [26]. All patients with relapsed disease activity were treated with high-dose steroids or immunosuppressive agents (azathioprine or methotrexate (MTX) and tocilizumab (TCZ)). Kaplan–Meier curves for relapse-free survival indicate significantly fewer relapses and a longer time to relapse in the GCA group than in the TA group (Figure 3).

### 3.4. Incidence of Cardiovascular Events in Patients with GCA and TA

There was no statistically significant difference in the incidence of cardiovascular events between the groups of patients with GCA and TA (Figure 4). Cardiovascular events occurred in 2 patients with GCA and 12 patients with TA. These cardiovascular events included cerebral aneurysms (1 GCA), internal carotid aneurysms (1 GCA), common carotid stenosis (2 TA), subclavian artery stenosis (2 TA), ascending aortic rupture (1 TA), chest tightness due to aortic regurgitation (1 TA), coronary artery stenosis (2 TA), back pain due to progressing aortitis (1 TA), stenosis of the superior iliac artery (1 TA), and femoral artery occlusion (1 TA).

### 3.5. Tapering of GC Dose

We evaluated the rates of patients who achieved prednisolone (PSL) tapering (PSL < 5.0 mg/day) in the GCA and TA groups (Figure 5). The number of patients who achieved steroid tapering was significantly higher in the GCA group than in the TA group (Figure 5).

### 3.6. GC Tapering According to the Presence of LVI or Concomitant Use of Immunosuppressant in GCA Patients

The 15 enrolled patients with GCA were divided into two groups according to the presence or absence of LVI: 7 patients in the LVI(+) group and 8 patients in the LVI(−) group. A comparison of the LVI(+) group and LVI(−) group characteristics is presented in Table 4. The LVI(+) group was significantly more likely to use TCZ for remission induction therapy than the LVI(−) group. The CRP level at diagnosis was significantly higher in the LVI(−) group. The GCA group was also divided into two groups according to the presence or absence of LVI and concomitant use of immunosuppressive drugs, and the number of patients who achieved steroid tapering (PSL < 5.0 mg/day) was evaluated. There was no significant difference in the rate of patients who achieved steroid tapering between the groups (Figure 6 and Figure 7).

## 4. Discussion

We analyzed Japanese patients’ clinical characteristics and outcomes with GCA in the present study. Our study revealed interesting findings regarding the prognosis of Japanese patients with GCA. Patients with GCA have been shown to have elderly onset disease and a worse prognosis than those with TA [27]. In our data, the onset age was consistent with the previous studies; however, the survival curve of the patients was comparable to that of patients with TA. Although there is evidence of an increased frequency of serious life-threatening vascular events in patients with GCA, the survival rates of patients with GCA are comparable to those of patients with TA [28]. Our results differ from a previous report of Japanese GCA patients in which the overall survival rates of GCA patients was significantly lower than that of TA patients [27]. However, other studies have demonstrated improved prognosis in patients with GCA, similar to TA, and the prognosis of GCA is still debated [4,29,30].

Maksimowicz-McKinnon et al. [31] reported that the frequency of aortic involvement in patients with GCA is 62%. In our data, aortic involvement was observed in approximately half of the patients with GCA; however, the rates of cardiovascular events in patients with GCA were similar to those in patients with TA. Increased deaths due to cardiovascular events, including ruptured aneurysms or aortic dissection, have been reported in patients with GCA and those with TA [12]. We observed an increased incidence of extracranial manifestations of GCA, such as LVI including aortic regions; however, our data showed that the survival rates of GCA are comparable to those of TA.

Previous studies have suggested that aortic involvement can be associated with a worse prognosis, and patients with GCA with LVI were associated with increased mortality [4]. Kermani et al. [4] showed that among patients with GCA, aortic manifestations were associated with increased mortality. On the other hand, Yamaguchi E. et al. reported that GCA without LVI has more active disease, more vascular damage, and worse survival than LV-GCA [32]. Our cohort’s overall survival of Japanese patients with GCA was favorable, and LVI was not associated with relapse or survival. Furthermore, the cumulative incidence of cardiovascular events in patients with GCA was similar to that of TA in our cohort study. Increased deaths due to cardiovascular events, including ruptured aneurysms or aortic dissection, have been demonstrated in patients with GCA. However, these events were rarely observed in cohort studies. Our study showed that the relapse-free survival rates of GCA were significantly higher than those of TA. In line with these clinical outcomes, the rates of patients with GCA who achieved steroid tapering (PSL < 5.0 mg/day) were significantly higher than those of patients with TA. These observations may account for the relatively short follow-up period in our cohort. Additionally, immunosuppressants or TCZ may account for a lower incidence of large-vessel events, such as the aneurysmal formation of dissections of large vessels. While the number of subjects was small, further prospective studies with larger scales and longer durations are needed to evaluate GCA disease outcomes.

MA González-Gay et al. [33] reported that in European countries, GCA patients with extracranial disease patterns in which LVI predominates are generally younger and have more features of polymyalgia rheumatica (PMR) than those with classic cranial disease patterns. However, our study showed no significant difference between patients with and without LVI. Recently, another cohort study of Japanese GCA patients reported no difference in age or prevalence of PMR with or without LVI [32]. Our study was consistent with the previous reports. Racial differences between European countries and Japan may be related to this result.

GCs remain the mainstay of treatment for GCA [6,7]. And GCs effectively control the clinical manifestations of GCA, whereas paradoxical GC treatment increases the risk of cardiovascular events by progressing the atherosclerotic process [3]. However, prospective studies assessing these essential issues are scarce. Recent EULAR recommendations for managing large vessel vasculitis advocate that the tapering of the GCs dose should maintain a target dose below 7.5 mg/day by combining a non-GCs immunosuppressive agent [22]. Regarding immunosuppressants, randomized controlled trials have shown that adding methotrexate is effective in patients with GCA, reducing the risk of flares and having a GCs-sparing effect [34]. A recent randomized controlled trial, Giant Cell Arteritis ACTEMRA (GiACTA), demonstrated that weekly treatment with tocilizumab in combination with GCs reduced disease activity and had a significant GCs-sparing effect in patients with GCA [9]. However, we could not demonstrate the GCs-sparing effects of these immunosuppressive treatments in patients with GCA. Further large-scale studies are needed to elucidate the role of these immunosuppressants on the GCs-sparing effect or clinical outcomes of GCA.

Our study has several limitations. This is a retrospective observational study with a small sample size. Furthermore, the median follow-up period for patients with GCA patients was only 25 months, despite the 10-year recruitment period. And we only evaluated the complications of the large-vessel disease, including large-artery stenosis and aortic aneurysm/dissection. The retrospective design of this study resulted in a lack of uniformity in immunosuppressive treatments, including PSL doses, and treatment decisions were left at the physician’s discretion. The follow-up periods were relatively limited to surveys of cardiovascular events that may occur several years after diagnosing large-vessel vasculitis.

## 5. Conclusions

Japanese patients who were GCA carriers may have a similar survival rate compared to those with TA. The relapse-free survival rates may be higher in patients with GCA than in patients with TA. However, large-scale prospective studies are required to confirm our results.

## Figures and Tables

**Figure 1 jpm-13-00387-f001:**
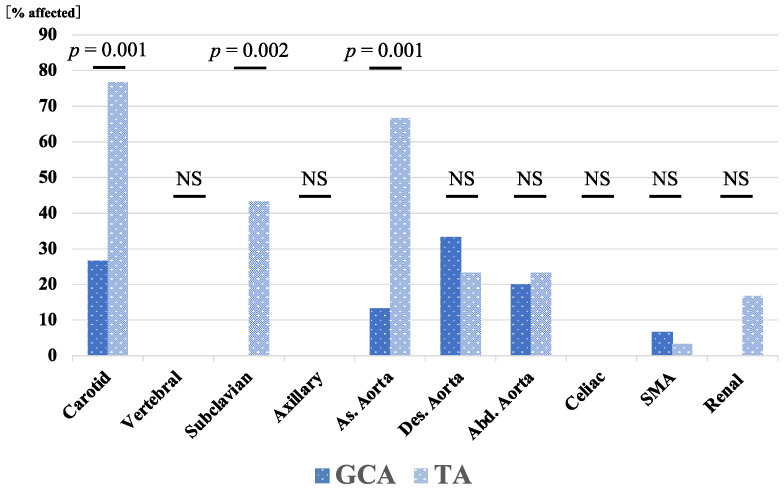
Distribution of arterial involvement in patients with GCA and TA. Carotid, subclavian, and ascending aorta involvement rates were significantly higher in patients with TA than in patients with GCA. As., ascending; Abd., abdominal; Des., descending; GCA, giant cell arteritis; NS, not significant; SMA, superior mesenteric artery; TA, Takayasu arteritis.

**Figure 2 jpm-13-00387-f002:**
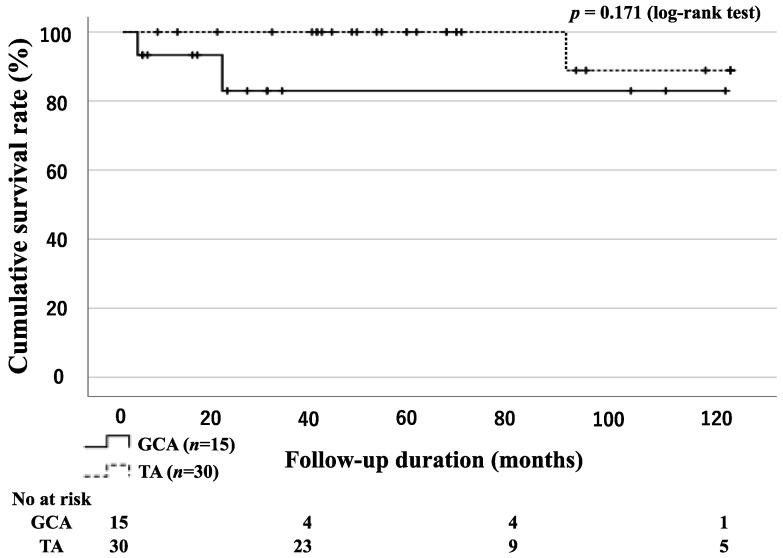
Survival curves in patients with GCA and TA. Kaplan–Meier curves showing the cumulative survival of patients with GCA and TA. No significant differences were observed between the GCA and TA groups. The starting point (0 months) was the date when observation began.

**Figure 3 jpm-13-00387-f003:**
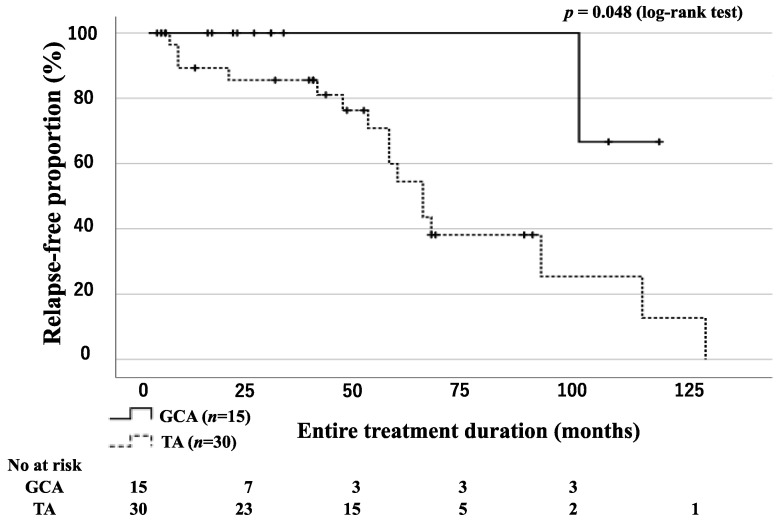
Relapse-free survival of patients with GCA and TA. Kaplan–Meier curves show relapse-free survival in patients with GCA (*n* = 15) and TA (*n* = 30). The starting point (0 months) was the date of the initiation of remission induction therapy. A significant difference was observed between the GCA and TA groups.

**Figure 4 jpm-13-00387-f004:**
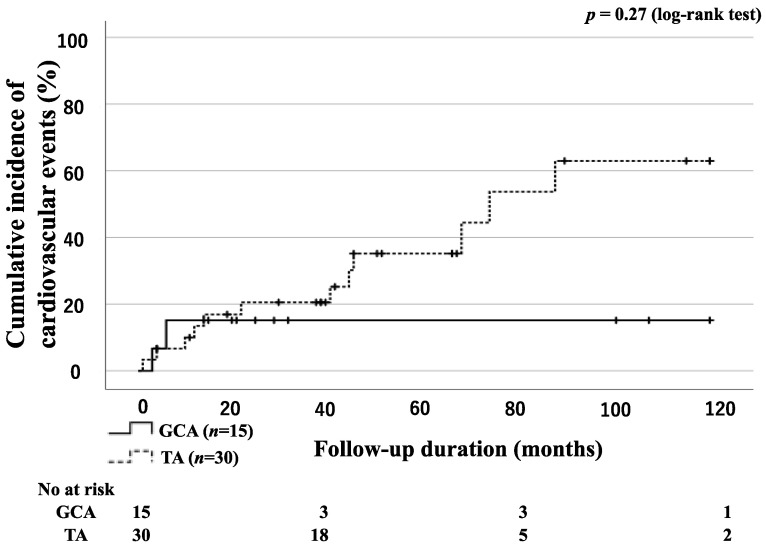
Cumulative incidence curves of cardiovascular events in patients with GCA and TA. Kaplan–Meier curves show the cumulative incidence of cardiovascular events in patients with GCA (*n* = 15) and TA (*n* = 30). No significant differences were observed between the GCA and TA groups. The starting point (0 months) was the date when observation began.

**Figure 5 jpm-13-00387-f005:**
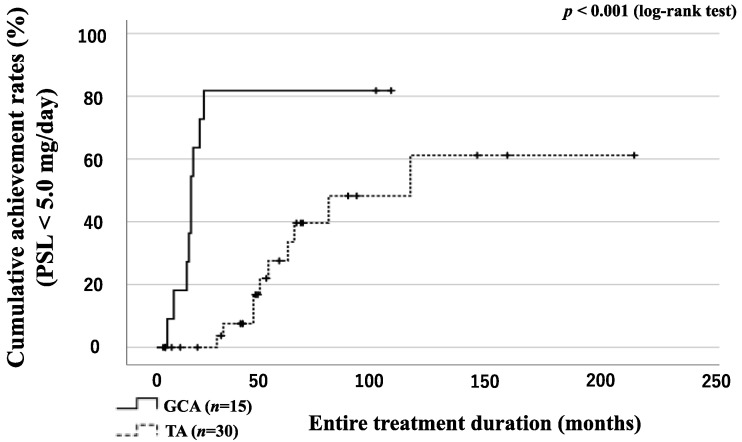
Rates of patients with GCA and TA achieving the tapered PSL dose (PSL < 5.0 mg/day). The proportions of patients achieving a tapered PSL dose (<5.0 mg/day) were compared between the GCA and TA groups using Kaplan–Meier curves. The starting point (0 months) was the date of the initiation of remission induction therapy. A significant difference was observed between the GCA and TA groups.

**Figure 6 jpm-13-00387-f006:**
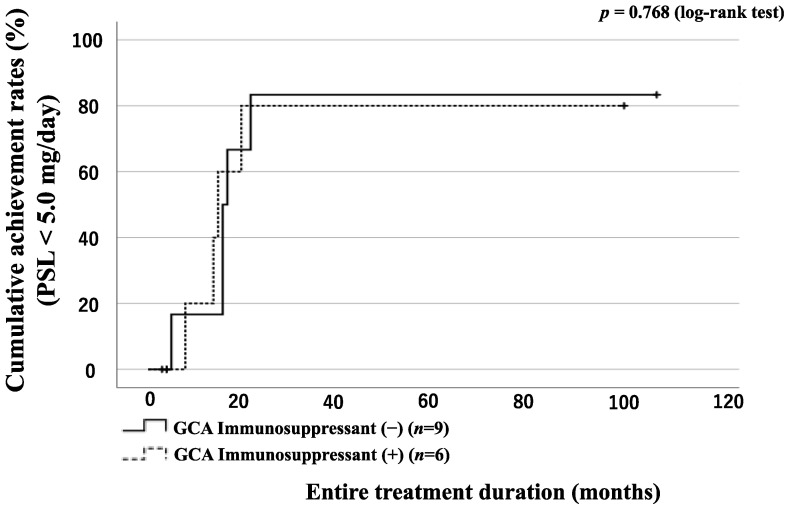
Rates of patients with GCA with and without immunosuppressant achieving the tapered PSL dose (PSL < 5.0 mg/day). Rates of patients with GCA achieving a tapered PSL dose (<5.0 mg/day). The proportions of patients who achieved a tapered PSL dose were compared with patients with GCA treated with or without immunosuppressants using Kaplan–Meier curves. The starting point (0 months) was the date of the initiation of remission induction therapy. There was no significant difference between the immunosuppressant (+) and immunosuppressant (−) groups.

**Figure 7 jpm-13-00387-f007:**
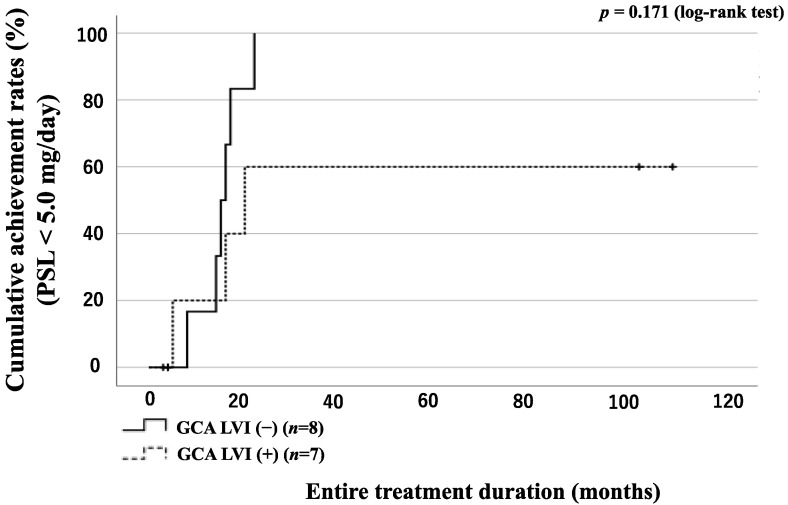
Rates of patients with GCA with and without LVI achieving the tapered PSL dose (PSL < 5.0 mg/day). Rates of patients with GCA achieving a tapered PSL dose (PSL < 5.0 mg/day). The proportions of patients who achieved a tapered PSL dose were compared with patients with GCA treated with or without LVI using Kaplan–Meier curves. The starting point (0 months) was the date of the initiation of remission induction therapy. There was no significant difference between the LVI(+) and LVI(−) groups.

**Table 1 jpm-13-00387-t001:** Comparisons of clinical features between patients with GCA and those with TA.

Characteristics	GCA (*n* = 15)	TA (*n* = 30)	*p*-Value
Male, *n* (%)	4 (26.7)	4 (13.3)	0.270
Age at onset (years), median (range)	77 (57–89)	24 (16–72)	<0.001 *
Follow-up period (months), median (range)	25 (3–121)	58 (7–122)	0.073
Entire treatment period (months), median (range)	25 (3-121)	58 (7-222)	0.073
Hypertension, *n* (%)	9 (60)	2 (6.7)	<0.001 *
Arterial stenosis, *n* (%)	0 (0)	8 (26.6)	0.027 *
Aneurysm, *n* (%)	2 (13.3)	2 (6.7)	0.459
Arterial involvement, *n* (%)	7 (46.7)	30 (100)	<0.001 *
Carotid artery, *n* (%)	4 (26.7)	23 (76.7)	0.001 *
Vertebral artery, *n* (%)	0 (0)	0 (0)	
Subclavian artery, *n* (%)	0 (0)	13 (43.3)	0.002 *
Axillary artery, *n* (%)	0 (0)	0 (0)	
Ascending aorta, *n* (%)	2 (13.3)	20 (66.7)	0.001 *
Descending aorta, *n* (%)	5 (33.3)	7 (23.3)	0.475
Abdominal aorta, *n* (%)	3 (20)	7 (23.3)	0.8
Celiac artery, *n* (%)	0 (0)	0 (0)	
Superior mesenteric artery, *n* (%)	1 (6.7)	1 (3.3)	0.609
Renal artery, *n* (%)	0 (0)	5 (16.7)	0.094
Initial dose of prednisolone (mg/day), median (range)	30 (10–40)	40 (0–60)	0.187
Steroid pulse therapy, *n* (%)	1 (6.7)	9 (30)	0.076
Autoimmune comorbidities, *n* (%)	1 (6.7)	6 (20)	0.245
Immunosuppressants, *n* (%)	2 (13.3)	7 (23.3)	0.429
TCZ, *n* (%)	4 (26.6)	15 (50)	0.135
Death, *n* (%)	2 (13.3)	1 (3.3)	0.205
ESR (mm/1^st^ hour), median (range)	65 (22-95)	56 (27–111)	0.488
CRP (mg/dl), median (range)	6.09 (0.33–21.31)	6.30 (0.2–35.09)	0.951
Relapse during the entire treatment period, *n* (%)	1 (6.7)	16 (53.3)	0.002 *
Relapse rate (per 100 person years), *n*	1.88	14.49	
Rate ratio, (95% CI)	6.78 (1.39-162.93)		0.013 *

GCA, giant cell arteritis; TA, Takayasu arteritis; TCZ, tocilizumab; ESR, erythrocyte sedimentation rate; CRP, C reactive protein; CI, confidence interval. * indicate a significant difference with *p* < 0.05.

**Table 2 jpm-13-00387-t002:** Other baseline characteristics of the enrolled GCA patients.

Characteristics	Value
Number, *n*	15
ACR criteria for active disease, *n* (%)	
Age at disease onset ≥ 50 years	15 (100)
New headache	15 (100)
Temporal artery abnormality	11 (73.3)
Elevated ESR	15 (100)
Abnormal temporal artery biopsy	4 (26.7)
PMR involvement, *n* (%)	7 (46.7)
Aches or pain in shoulders, *n*/7 (%)	5/7 (71.4)
Aches or pain in the neck, upper arms, buttocks, hips, or thighs, *n* (%)	6/7 (85.7)
Stiffness in affected areas, *n* (%)	5/7 (71.4)
Limited range of motion in affected areas, *n* (%)	2/7 (28.6)
Mild fever, *n* (%)	5/7 (71.4)
Fatigue, *n* (%)	4/7 (57.1)
Unintended weight loss, *n* (%)	6/7 (85.7)
Depression, *n* (%)	4/7 (57.1)
Ischemic optic neuropathy, *n* (%)	0 (0)
Aortic regurgitation, *n* (%)	1 (6.7)
Malignancy, *n* (%)	2 (13.3)

GCA, giant cell arteritis; ACR, American College of Rheumatology; ESR, erythrocyte sedimentation rate; PMR, polymyalgia rheumatica.

**Table 3 jpm-13-00387-t003:** Other baseline characteristics of the enrolled TA patients.

Characteristics	Value
Number, *n*	30
HLA typing	
HLA-B52 positive, *n*/patients HLA tested (%)	7/25 (28.0)
Classification of Takayasu arteritis, *n* (%)	
I	8 (26.7)
IIA	6 (20)
IIB	6 (20)
III	0 (0)
IV	0 (0)
V	10 (33.3)

TA, Takayasu arteritis; HLA, human leukocyte antigen.

**Table 4 jpm-13-00387-t004:** Comparisons of clinical features between LVI(+) and LVI(−) groups.

Characteristics	LVI(+) (*n* = 7)	LVI(−) (*n* = 8)	*p*-Value
Male, *n* (%)	2 (28.6)	2 (25.0)	0.876
Age at onset (years), median (range)	80 (57–89)	76 (65–86)	0.598
Entire treatment period (months), median (range)	29 (15–121)	17 (3–109)	0.619
ACR criteria for active disease			
Age at disease onset ≧ 50 years	7 (100)	8 (100)	
New headache	7 (100)	8 (100)	
Temporal artery abnormality	4 (57.1)	7 (87.5)	0.185
Elevated ESR	7 (100)	8 (100)	
Abnormal temporal artery biopsy	2 (28.6)	2 (25.0)	0.876
PMR, *n* (%)	2 (28.6)	5 (62.5)	0.189
Hypertension, *n* (%)	6 (85.7)	3 (37.5)	0.057
Malignancy, *n* (%)	0 (0)	2 (25.0)	0.155
Steroid pulse therapy, *n* (%)	1 (14.3)	0 (0)	0.268
Initial dose of prednisolone (mg/day), median (range)	30 (20–40)	30 (10–40)	0.875
Immunosuppressants, *n* (%)	0 (0)	2 (25.0)	0.155
TCZ, *n* (%)	4 (57.1)	0 (0)	0.013 *
Death, *n* (%)	0 (0)	2 (25.0)	0.155
Relapse, *n* (%)	0 (0)	1 (12.5)	0.333
ESR (mm/1st hour), median (range)	59 (22–95)	70.5 (36–88)	0.557
CRP (mg/dl), median (range)	3.77 (0.99–9.76)	13.3 (0.3321.34)	0.009 *

LVI, large vessel involvement; PMR, polymyalgia rheumatica; TCZ, tocilizumab; ESR, erythrocyte sedimentation rate; CRP, C reactive protein. * indicate a significant difference with *p* < 0.05.

## Data Availability

The raw data supporting the conclusions of this article will be made available by the authors, without undue reservation.

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
