# Peer review of "Clinical Features and Outcomes of Japanese Patients with Giant Cell Arteritis: A Comparison with Takayasu Arteritis"

_jpm, 2023, doi:10.3390/jpm13030387_

Round 1
Reviewer 1 Report
The authors evaluated the differences between giant cell arteritis (GCA) and Takayasu's arteritis (TAK) in the Japanese population. Given that GCA is a rare vasculitis in Asian countries, the comparison between the two vasculitides made by the authors is of potential interest as it provides useful information.
The possible limitations elegantly listed by the authors do not detract from the manuscript. However, I suggest that the authors make some changes/comments that in no way belittle the relevance of the report.
Introduction:
Replace reference 3 with a more recent one that addresses the same issue (PMID: 15525845 DOI: 10.1097/01.md.0000145366.40805.f8)
Lines 43-44: Rewrite de following paragraph “The efficacy of tocilizumab has been demonstrated in a recent randomized controlled trial, suggesting steroid-sparing effects and the induction of clinical remission [9]” as follows: “The efficacy of anti-IL-6 receptor tocilizumab in GCA to reduce the accumulated dose of glucocorticoids and the risk of relapse has been demonstrated both in clinical trials and in several studies based on daily clinical practice.” [9, AND PMID: 25697557 DOI: 10.1016/j.semarthrit.2014.12.005, AND PMID: 30655091 DOI: 10.1016/j.semarthrit.2019.01.003]
Lines 50-52: “an increased risk of potentially life-threatening ischemic or cardiovascular events has been suggested during the therapeutic disease course [11-14]” Replace any of the ref. 11 or ref. 12 and add one related to cerebrovascular accidents that have been observed in some GCA patients at the time of GCA diagnosis or within the first few months after initiation of glucocorticoid therapy [PMID: 19593228 DOI: 10.1097/MD.0b013e3181af4518}
Lines 55-59 (last paragraph of the Introduction): REPLACE “We conducted a retrospective multicenter study of 15 patients with GCA to evaluate long-term clinical outcomes and prognosis. The present study aimed to analyze GCA’s clinical features and outcomes to determine its prognostic significance in Japanese patients. We also compared the long-term outcomes of patients with GCA with those with TA.” WITH the following sentence “Since GCA is a rare vasculitis in Asian countries [Ref. PMID: 11723761 DOI: 10.1016/s0889-857x(05)70232-5 ], information regarding differences in outcome between GCA and TAK is scarce in this part of the world. Because of this, we established a retrospective study of GCA and TAK patients seen at our center over a 10-year period.”
Results:
Please, add units; ESR mm/1st hour and CRP mg/dl.
As the authors noted, it is very difficult to draw conclusions about survival, since GCA and TAK have a very different age of onset. For example, hypertension is more common in older people and this may be the reason why it is seen more frequently in patients with GCA.
Relapses: One comment: I would have expected a higher number of relapses in the GCA group; consider commenting on that finding in the Discussion.
Lines 167-168: “back pain due to processing aortitis”- It could be "progressing" (instead of processing)”?
Did you perform a temporal artery biopsy on patients with GCA? I assume that in those with predominant extracranial LVI the result was negative.
Discussion:
Regarding LVI in GCA patients: In European countries, GCA patients with an extracranial disease pattern manifested by predominant LVI are generally younger and have more common PMR features than those with the classic cranial disease pattern (Ref. PMID: 31810548 DOI: 10.1016/ j.ber.2019.06.006). However, this was not the case in this cohort of Japanese GCA patients. Make a comment on that.
Author Response
Response to reviewer 1
The authors evaluated the differences between giant cell arteritis (GCA) and Takayasu's arteritis (TAK) in the Japanese population. Given that GCA is a rare vasculitis in Asian countries, the comparison between the two vasculitides made by the authors is of potential interest as it provides useful information.
The possible limitations elegantly listed by the authors do not detract from the manuscript. However, I suggest that the authors make some changes/comments that in no way belittle the relevance of the report.
We wish to express our appreciation for your important comments on our manuscript. We have added more detailed information to the manuscript. We have made some necessary changes to present the more obscure items. The comments have helped us to improve the quality of our manuscript significantly.
Introduction:
Replace reference 3 with a more recent one that addresses the same issue (PMID: 15525845 DOI: 10.1097/01.md.0000145366.40805.f8)
We appreciated your comment. We have replaced the reference you indicated.
Lines 43-44: Rewrite de following paragraph “The efficacy of tocilizumab has been demonstrated in a recent randomized controlled trial, suggesting steroid-sparing effects and the induction of clinical remission [9]” as follows: “The efficacy of anti-IL-6 receptor tocilizumab in GCA to reduce the accumulated dose of glucocorticoids and the risk of relapse has been demonstrated both in clinical trials and in several studies based on daily clinical practice.” [9, AND PMID: 25697557 DOI: 10.1016/j.semarthrit.2014.12.005, AND PMID: 30655091 DOI: 10.1016/j.semarthrit.2019.01.003]
We appreciated your pertinent comment. We have rewritten the text as you indicated.
Lines 50-52: “an increased risk of potentially life-threatening ischemic or cardiovascular events has been suggested during the therapeutic disease course [11-14]” Replace any of the ref. 11 or ref. 12 and add one related to cerebrovascular accidents that have been observed in some GCA patients at the time of GCA diagnosis or within the first few months after initiation of glucocorticoid therapy [PMID: 19593228 DOI: 10.1097/MD.0b013e3181af4518}
We appreciate for your comments. We have revised the text as you indicated.
Lines 55-59 (last paragraph of the Introduction): REPLACE “We conducted a retrospective multicenter study of 15 patients with GCA to evaluate long-term clinical outcomes and prognosis. The present study aimed to analyze GCA’s clinical features and outcomes to determine its prognostic significance in Japanese patients. We also compared the long-term outcomes of patients with GCA with those with TA.” WITH the following sentence “Since GCA is a rare vasculitis in Asian countries [Ref. PMID: 11723761 DOI: 10.1016/s0889-857x(05)70232-5 ], information regarding differences in outcome between GCA and TAK is scarce in this part of the world. Because of this, we established a retrospective study of GCA and TAK patients seen at our center over a 10-year period.”
Thank you for your significant comment. We have made some changes to your text and revised the manuscript. This important point has been added to the manuscript on page 2, lines 59-60.
“Since GCA is a rare vasculitis in Asian countries [18], information regarding the differences in clinical outcome between GCA and TA is scarce in real-world.”
Results:
Please, add units; ESR mm/1st hour and CRP mg/dl.
We appreciated your pertinent comment. As you indicated, we have added a description of the units.
As the authors noted, it is very difficult to draw conclusions about survival, since GCA and TAK have a very different age of onset. For example, hypertension is more common in older people and this may be the reason why it is seen more frequently in patients with GCA.
Thank you for your pertinent comment. We also believe that age may be a factor in why hypertension is more common in the GCA group than in the TA group. We have added to the manuscript on this, which can be found on lines 138-139, page 3.
“Similarly, the prevalence of hypertension was significantly higher in patients with GCA than in those with TA (p <0.001).”
Relapses: One comment: I would have expected a higher number of relapses in the GCA group; consider commenting on that finding in the Discussion.
We appreciate for your critical comment. There was only one relapse of GCA in this study. This may be related to the short follow-up period, as mentioned in the study limitations. However, with regard to the relapse rate per person-year, the GCA group also had fewer relapses. TCZ has been shown to suppress GCA relapse and preserve GC (Ref. PMID: 28745999 DOI: 10.1056/NEJMoa1613849). The high number of patients with LVI receiving induction remission therapy with TCZ and GC and the short duration of treatment probably contributed to the low number of relapses. We described the issue in limitation of the revised manuscript, which can be found on lines 361-362, page 11.
“Furthermore, the median follow-up period for patients with GCA patients was only 25 months, despite the 10-year recruitment period.”
Lines 167-168: “back pain due to processing aortitis”- It could be "progressing" (instead of processing)”?
We appreciate for your comment. "progressing" is correct. We have revised the manuscript.
Did you perform a temporal artery biopsy on patients with GCA? I assume that in those with predominant extracranial LVI the result was negative.
Thank you for your significant comment. In this study, there were 4 cases in the GCA group in which multinucleated giant cells were proven by temporal artery biopsy (2 in the LVI(+) group and 2 in the LVI(-) group). Overall, 6/15 patients underwent temporal artery biopsy (3 in the LVI(+) group and 3 in the LVI(-) group). Thus, the results did not differ significantly between the LVI(-) and LVI(+) groups.
Discussion:
Regarding LVI in GCA patients: In European countries, GCA patients with an extracranial disease pattern manifested by predominant LVI are generally younger and have more common PMR features than those with the classic cranial disease pattern (Ref. PMID: 31810548 DOI: 10.1016/ j.ber.2019.06.006). However, this was not the case in this cohort of Japanese GCA patients. Make a comment on that.
We appreciated your comment. It has recently been reported that in a cohort study of 36 Japanese GCA patients, there was no difference in age or prevalence of PMR between groups with and without LVI (Ref. PMID: 35141755 DOI: 10.1093/mr/roac013). Our study was consistent with previous reports. Racial differences between European countries and Japan may be responsible for this result. We have added to the manuscript on this, which can be found on lines 337-343, page 10.
“MA González-Gay et al. [33] reported that in European countries, GCA patients with extracranial disease patterns in which LVI predominates are generally younger and have more features of polymyalgia rheumatica (PMR) than those with classic cranial disease patterns. However, our study showed no significant difference between patients with and without LVI. Recently, an-other cohort study of Japanese GCA patients reported no difference in age or prevalence of PMR with or without LVI [34]. Our study was consistent with previous reports. Racial differences between European countries and Japan may be related to this result.”
Reviewer 2 Report
Thank you for inviting my review of this paper which compares findings and outcomes between 15 Japanese patients with GCA and 30 with TA. The authors state that they showed no difference in survival but relapse-free survival was better for GCA patients who also achieved greater dosage reductions in prednisone. There are a few points that merit consideration:
MAJOR
1 The authors describe enrolling consecutive patients over a 10 year period at multiple centres, yet only identified 15 GCA patients. Were any exclusion criteria applied or does this represent a very low prevalence rate of GCA in Japan? If the latter, further discussion and referencing would be useful.
2 By contrast, the authors enrolled 30 TA patients over the same time period. This feels much more akin to that which might be expected for the total population served over a decade. Is there data on the prevalence of TA in Japan? Is TA really twice as common as GCA in Japan?
3 In addition, mean followup of the GCA patients was only 25 months despite a 10 year recruitment window. By comparison,mean followup for TA patients was 58 months. The shorter follow-up for GCA reduces the validity of the comparison between the two conditions for all the variables reported
4 Furthermore,the very low numbers and short follow up of just 15 GCA patients makes all conclusions (and comparisons) very limited. Just one more relapse over a mean difference of 33 months of follow up would increase the relapse rate by 7%.A more meaningful analysis would be to calculate relapse rate by patient /year of follow up which would reduce the apparent difference between the two conditions.
5 The initial dose of prednisone for GCA patients appears lower than usual at a mean of 30mg with some patients getting as little as 10mg and none over 40mg. This contrasts with data and recommendations elsewhere. Can the authors explain and discuss this?
MINOR
1 Table 3 could usefully be expanded to incorporate the comparators shown in Tables 1 and 2, rendering them unnecessary
2 Figure 3 appears to suggest that no TA patients were relapse free but the data shows that 14 /30 were free of relapse through follow up. The axis needs to be relabelled or the data replotted.
3 Lines 11-2 describe the significant age difference between the two patient groups but then reiterate this.The second sentence can be incorporated by stating the P value of the comparison in the first sentence.
4 The Abstract should provide the age and sex data for the TA group as well as the GCA patients
5 Isn't the difference in hypertension a feature of age and higher steroid dosage for the TA group,rather than due to the disease process itself?
6 In lines 118-20, both groups are stated to have greater carotid disease but the TA group which have much more than the GCA patients
Author Response
Response to reviewer 2
Thank you for inviting my review of this paper which compares findings and outcomes between 15 Japanese patients with GCA and 30 with TA. The authors state that they showed no difference in survival but relapse-free survival was better for GCA patients who also achieved greater dosage reductions in prednisone. There are a few points that merit consideration:
We wish to express our appreciation to your insightful comments on our paper. The comments have helped us significantly improve the paper.
MAJOR
1 The authors describe enrolling consecutive patients over a 10 year period at multiple centres, yet only identified 15 GCA patients. Were any exclusion criteria applied or does this represent a very low prevalence rate of GCA in Japan? If the latter, further discussion and referencing would be useful.
Thank you for your constructive comments. We recruit all our patients and do not exclude them. GCA is rare in Asians, and the small number of patients in this study may represent the very low prevalence of GCA in Japan. In fact, other reviewers made the same point. This important point has been added to the manuscript on page 2, lines 59-60.
“Since GCA is a rare vasculitis in Asian countries [18], information regarding the differences in clinical outcome between GCA and TA is scarce in real-world.”
2 By contrast, the authors enrolled 30 TA patients over the same time period. This feels much more akin to that which might be expected for the total population served over a decade. Is there data on the prevalence of TA in Japan? Is TA really twice as common as GCA in Japan?
We appreciated your comment. A 2017 Japanese epidemiological study estimated the number of clinically diagnosed TA and GCA patients to be 5,320 (95% confidence interval [CI], 4,810-5,820) and 3,200 (95% CI, 2,830-3,570), respectively (Ref. PMID: 36737863 DOI: 10.1093/mr/road019). We described that information in the revised manuscript, which can be found on lines 60-63, page 2.
“A 2017 Japanese epidemiological study estimated the number of clinically diagnosed TA and GCA patients to be 5,320 (95% confidence interval [CI], 4,810-5,820) and 3,200 (95% CI, 2,830-3,570), respectively [19]. This is an extremely small number in relation to the Japanese population.”
3 In addition, mean followup of the GCA patients was only 25 months despite a 10 year recruitment window. By comparison,mean followup for TA patients was 58 months. The shorter follow-up for GCA reduces the validity of the comparison between the two conditions for all the variables reported.
Thank you for your important comment. The median follow-up period for the GCA group was 25 months due to the presence of several patients who had recently started treatment. We described the issue in limitation of the revised manuscript, which can be found on lines 361-362, page 11.
“Furthermore, the median follow-up period for patients with GCA patients was only 25 months, despite the 10-year recruitment period.”
4 Furthermore,the very low numbers and short follow up of just 15 GCA patients makes all conclusions (and comparisons) very limited. Just one more relapse over a mean difference of 33 months of follow up would increase the relapse rate by 7%.A more meaningful analysis would be to calculate relapse rate by patient /year of follow up which would reduce the apparent difference between the two conditions.
Thank you for your important feedback. Relapses per person-year and the ratio were calculated. The results showed that TA had more relapses per person-year. We have added to the manuscript on this, which can be found on lines 142-144, page 3.
“The relapse rate ratio for TA group to GCA group was 6.78, indicating that TA group is more likely to relapse than GCA group (95% CI 1.39–162.93, p=0.013).”
5 The initial dose of prednisone for GCA patients appears lower than usual at a mean of 30mg with some patients getting as little as 10mg and none over 40mg. This contrasts with data and recommendations elsewhere. Can the authors explain and discuss this?
Thank you for your query. One possible reason for the lower PSL dosage in GCA group in this study could be that most patients with GCA were elderly and had more comorbidities, such as hypertension and type 2 diabetes, which may have reduced PSL dosage. In addition, there were no GCA patients with ocular arterial lesions in this study. Thus, they may have received moderate doses of steroids.
MINOR
1 Table 3 could usefully be expanded to incorporate the comparators shown in Tables 1 and 2, rendering them unnecessary.
Thank you for your comments. We have summarized in Table 1 all the comparable baseline characteristics of the patients in both groups. Items that should be described individually are summarized in Tables 2 and 3, respectively. We believe that the tables are now more clear and concise.
2 Figure 3 appears to suggest that no TA patients were relapse free but the data shows that 14 /30 were free of relapse through follow up. The axis needs to be relabelled or the data replotted.
Thank you for your comments. The Kaplan-Meier curve for the entire treatment period in this study is shown in the figure 3. Indeed, there were 14/30 relapse-free cases in the TA group. There are cases in which the total treatment period is 10 years. However, most of the cases were treated for about 5 years under this Kaplan-Meier curve, and a significant number of them were terminated. Therefore, it appears that there are no relapse-free cases because the Kaplan-Meier curves are terminated in the cases that have not relapsed. In fact, there are relapse-free cases, but the Kaplan-Meier curve seems to make it appear that relapse-free cases do not exist.
3 Lines 11-2 describe the significant age difference between the two patient groups but then reiterate this.The second sentence can be incorporated by stating the P value of the comparison in the first sentence.
We appreciated your pertinent comment. We have rewritten the text as you indicated.
4 The Abstract should provide the age and sex data for the TA group as well as the GCA patients
We appreciate for your comments. We have revised the text as you indicated.
5 Isn't the difference in hypertension a feature of age and higher steroid dosage for the TA group,rather than due to the disease process itself?
Thank you for your comment. We also believe that age may be a factor in why hypertension is more common in the GCA group than in the TA group. We have added to the manuscript on this, which can be found on lines 138-139, page 3.
“Similarly, the prevalence of hypertension was significantly higher in patients with GCA than in those with TA (p <0.001).”
6 In lines 118-20, both groups are stated to have greater carotid disease but the TA group which have much more than the GCA patients
Thank you for your important comment. We have changed the manuscript to a more appropriate wording. We have revised the manuscript on this, which can be found on lines 145-148, page 3.
“Aortic lesions were found primarily in the carotid artery, descending aorta, and ab-dominal aorta in patients with GCA. However, the carotid, subclavian, and ascending aorta lesion rates were significantly higher in patients with TA than patients with GCA (Figure 1).”
Round 2
Reviewer 2 Report
Thank you for addressing all my concerns in sufficient detail although I continue to wonder whether the small numbers and reduced follow up times in the GCA group might influence the relapse rate ratio.